# Systemic Sclerosis with Interstitial Lung Disease: Identification of Novel Immunogenetic Markers and Ethnic Specificity in Kazakh Patients

**DOI:** 10.3390/epidemiologia6030041

**Published:** 2025-08-06

**Authors:** Lina Zaripova, Abay Baigenzhin, Zhanar Zarkumova, Zhanna Zhabakova, Alyona Boltanova, Maxim Solomadin, Alexey Pak

**Affiliations:** 1Department of Internal Medicine, JSC National Scientific Medical Center, Astana Medical University, 42 Abylai Khan Ave., Astana 010009, Kazakhstan; l.zaripova@nnmc.kz (L.Z.); a.baigenzhin@nnmc.kz (A.B.); zhanna_zhabakova@mail.ru (Z.Z.); alyona.boltanova@gmail.com (A.B.); maks50@gmail.com (M.S.); bofur@rambler.ru (A.P.); 2School of Residency, Department of Internal Medicine No. 2, NCJSC Astana Medical University, 49/A Beibitshilik St., Astana 010000, Kazakhstan

**Keywords:** systemic sclerosis, interstitial lung disease, biomarkers, genetic polymorphisms, Kazakh population

## Abstract

Systemic sclerosis (SSc) is an autoimmune connective tissue disorder characterized by vascular abnormalities, immune dysfunction, and progressive fibrosis. One of the most common manifestations of SSc is interstitial lung disease (ILD), known by a progressive course leading to significant morbidity and mortality. **Aim:** to investigate autoantibodies, cytokines, and genetic markers in SSc-ILD through a systematic review and analysis of a Kazakh cohort of SSc-ILD patients. **Methods:** A PubMed search over the past 10 years was performed with “SSc-ILD”, “autoantibodies”, “cytokines”, and “genes”. Thirty patients with SSc were assessed for lung involvement, EScSG score, and modified Rodnan skin score. IL-6 was measured by ELISA, antinuclear factor on HEp-2 cells by indirect immunofluorescence, and specific autoantibodies by immunoblotting. Genetic analysis was performed using a 120-gene AmpliSeq panel on the Ion Proton platform. **Results:** The literature review identified 361 articles, 26 addressed autoantibodies, 20 genetic variants, and 12 cytokine profiles. Elevated levels of IL-6, TGF-β, IL-33, and TNF-α were linked to SSc. Based on the results of the systemic review, we created a preliminary immunogenic panel for SSc-ILD with following analysis in Kazakh patients with SSc (*n* = 30). Fourteen of them (46.7%) demonstrated signs of ILD and/or lung hypertension, with frequent detection of antibodies such as Scl-70, U1-snRNP, SS-A, and genetic variants in SAMD9L, REL, IRAK1, LY96, IL6R, ITGA2B, AIRE, TREX1, and CD40 genes. **Conclusions:** Current research confirmed the presence of the broad range of autoantibodies and variations in IRAK1, TNFAIP3, SAMD9L, REL, IRAK1, LY96, IL6R, ITGA2B, AIRE, TREX1, CD40 genes in of Kazakhstani cohort of SSc-ILD patients.

## 1. Introduction

Systemic sclerosis (SSc) is a connective tissue disease (CTD) marked by fibrosis affecting the skin and internal organs, including lungs, kidneys, and gastrointestinal tract. Damage to microvasculature occurs via activation of the immune response, producing antibodies [1]. Interstitial lung disease (ILD) is characterized by inflammation and fibrosis of the lungs, leading to severe respiratory failure [2]. SSc-ILD is more commonly observed in patients with diffuse cutaneous systemic sclerosis (dcSSc), which shows the highest incidence of clinically significant ILD, with a rate reaching 86.1% [3]. The extent and severity of skin and lung involvement in SSc do not directly correlate. However, various patterns of lung involvement have been observed in different SSc subgroups. Patients with diffuse SSc are more likely to have restrictive lung disease, while those with limited SSc tend to have a higher prevalence of pulmonary hypertension [4].

The role of inflammation and the specific mechanisms of fibroblast activation in systemic sclerosis-related interstitial lung disease (SSc-ILD) remains an ongoing area of research. Inflammation is one of the initial steps in the development of SSc-ILD, leading to the recruitment of immune cells into the interstitial and alveolar compartments. This process results in epithelial cell damage, and research suggests that the degree of alveolar epithelial injury is a significant factor in determining the likelihood of disease progression [5]. Even minor epithelial damage may contribute to increased susceptibility to more severe lung complications. According to experimental models, intratracheal instillation of mildly acidic unbuffered saline induces lung injury and significant fibrosis [5,6].

Another theory of the development of ILD in patients with SSc is based on the impact of chronic aspiration. This theory suggests that gastroesophageal reflux (GER) can lead to ILD, which is supported by radiological and histological data, using pH-impedance monitoring methods [7,8,9]. A potential association between the severity of esophageal motor dysfunction and the presence of ILD, as detected by pulmonary function tests (PFT) and high-resolution computed tomography (HRCT), was investigated in patients with systemic sclerosis [10]. The findings suggest that GER may be a contributing factor in the development of ILD in SSc. Furthermore, patients with more severe esophageal involvement may benefit from more frequent monitoring of pulmonary function, and early and proactive management of GER may help slow down the progression of lung disease in SSc patients [10].

The serological profile in SSc patients plays a significant role in the prognosis of the current ILD: anti-topoisomerase I (ATA; anti-Scl-70) antibodies [11,12] are predominantly associated with the development of pulmonary fibrosis, while anti-U3 RNP antibodies are linked to a higher incidence of pulmonary hypertension and cardiac involvement [3]. In contrast, anti-centromere antibodies (ACA) are rarely found in patients with SSc-ILD [13].

Histopathologic and radiographic evaluations indicate that SSc-ILD most often presents with a nonspecific interstitial pneumonia pattern, which is observed in up to 78% of cases. This is followed by usual interstitial pneumonia in up to 36% of cases, while other patterns such as organizing pneumonia are comparatively uncommon [14,15,16]. Pulmonary complications are a significant cause of morbidity and mortality in connective tissue diseases (CTD), and in some cases, such as SSc, they can affect up to half of individuals with more extensive forms of extra-pulmonary disease [17]. Pulmonary fibrosis and pulmonary arterial hypertension (PAH) are the most common lung conditions linked to SSc, which mainly cause deaths related to SSc [18,19]. ILD is observed in CTDs with the following median prevalence rates: SSc—47%, polymyositis/dermatomyositis—41%, mixed connective tissue disease—56%, and others [17]. The reported prevalence of ILD was identified as the leading cause of death, accounting for 17% of cases, followed closely by PAH, which contributed to 15% of deaths [11,20]. In the largest observational study conducted to date, SSc-ILD ranges from 25% to 90%, depending on the diagnostic methods used and the criteria applied to define the disease [21,22,23].

## 2. Materials and Methods

Possible genetic mutations, autoantibodies, and cytokines involved in autoimmune aggression in SSc identified through the review of relevant literature from biomedical databases, as well as by comparing with genetic panels offered by leading manufacturers. The literature search was performed on PubMed with the combination of keywords, including “SSc-ILD”, “genetic mutations”, “autoantibodies”, and “cytokines” for the last 10 years. Full-text articles relevant to these topics were included in the review. Priority was assigned to peer-reviewed research articles, systematic reviews, and meta-analyses. Furthermore, the Ion AmpliSeq™ Designer gene database was employed to pinpoint genes linked to specific diseases.

### 2.1. Study Population

This study initially included 30 adult individuals of Kazakh ethnicity diagnosed with the diffuse cutaneous form of systemic sclerosis (SSc), in accordance with the 2013 ACR/EULAR classification criteria. Participants were enrolled from the outpatient and inpatient departments of the National Scientific Medical Center between August 2023 and June 2024. SSc patients were recruited and examined for lung injury, including ILD and lung hypertension. The European Scleroderma Study Group (EScSG) score and skin impairment by Rodnan were assessed. Among these 30 SSc patients, fourteen (46.7%) demonstrated signs of ILD and/or lung hypertension. In addition, the control group (n = 18) consisting of 18 age- and sex-matched healthy Kazakh individuals, with no history of autoimmune or connective tissue disorders, was assembled through routine health screenings and outreach programs. This control group was used in part to compare some characteristics like gene expression to the patients’ group.

### 2.2. Ethical Considerations

The research protocol was approved by the Institutional Human Ethics Committee of the National Scientific Medical Center (Protocol No. 085/KI-79). The study was conducted in accordance with the principles of the Declaration of Helsinki. All participants were informed of the objectives and procedures of the study and gave written informed consent. Anonymized identifiers were assigned to all participants to preserve confidentiality. Only non-identifiable information, such as diagnosis and sex, were recorded.

### 2.3. Sample Collection and Preparation

Venous blood samples were collected from both patients and controls. Routine laboratory tests were performed immediately upon delivery to the central laboratory. For genetic and immunological analyses, aliquots were stored under conditions that ensured the preservation of molecular integrity.

### 2.4. Autoantibody Profiling

The detection of antinuclear factor was estimated by human laryngeal carcinoma cell line HEp-2 cells due to the manufacture protocol detected by indirect immunofluorescence (IIF) assay with semi-quantitative (titer) determination of immunoglobulin G (IgG) antibodies to nuclear and cytoplasmic antigens in blood serum. The IIF procedure was performed with HEp-2 cell substrates using ANA Plus and ANA Cytobead kits on the Aklides automated platform (Medipan GmbH, Dahlewitz, Germany). Fluorescein-labeled secondary antibodies were applied to detect bound autoantibodies, with fluorescence patterns evaluated following the International Consensus on ANA Patterns (ICAP) classification.

Antibodies to Scl-70 (anti-topoisomerase I antibodies), U1-snRNP (antibodies to U1 small nuclear ribonucleoprotein), CENP-B (antibodies to centromere protein B), SS-A/Ro52 (antibodies to the 52 kDa Ro/SSA antigen), SS-A/Ro60 (antibodies to the 60 kDa Ro/SSA antigen), Sm/RNP (antibodies to Smith/ribonucleoprotein complex), Sm (antibodies to Smith antigen), SS-B (antibodies to La antigen), and Rib-P0 (antibodies to ribosomal phosphoprotein P0), and Nucleosome were determined by immunoblotting. For extended antibody profiling, immunoblotting was employed using commercially available ANA-12 line blot assays. Sera were incubated with antigen-coated strips, and antibody binding was visualized using enzyme-conjugated secondary antibodies and a chromogenic substrate. Positive results from the line blot were confirmed and quantified using Cytobead technology.

### 2.5. Cytokine Quantification

Serum levels of interleukin-6 (IL-6) were measured using enzyme-linked immunosorbent assay (ELISA) kits. The assays were conducted on an Alisei analyzer (Radim Diagnostics, Freiburg, Germany) according to the manufacturer’s protocols, employing monoclonal and polyclonal antibodies specific to IL-6. Optical density was measured spectrophotometrically at 450 nm, and concentrations were calculated based on standard calibration curves.

### 2.6. Genetic Analysis

Genomic DNA was extracted from whole blood samples using a silica column-based purification kit. DNA concentration and purity were verified by fluorometric quantification. A custom autoimmune gene panel comprising 120 target genes associated with systemic sclerosis and other autoimmune diseases was used for targeted next-generation sequencing (NGS).

Target regions were amplified via multiplex PCR, and unique barcodes were assigned to each sample. Libraries underwent magnetic bead-based purification and were sequenced using the Ion Proton platform (Thermo Fisher Scientific, Asheville, NC, USA). The sequencing output was subjected to bioinformatics analysis for quality control, variant calling, and annotation. Variants were filtered based on coverage depth, predicted functional impact, and association with autoimmune diseases, utilizing databases such as ClinVar and dbSNP. ClinVar (NCBI, Bethesda, MD, USA) provides clinical interpretations of genetic variants (https://www.ncbi.nlm.nih.gov/clinvar/), (accessed on 18 July 2025) while dbSNP (NCBI, USA) catalogs genetic polymorphisms and variants (https://www.ncbi.nlm.nih.gov/snp/) (accessed on 18 July 2025).

### 2.7. Statistical Analysis

Statistical analyses were conducted utilizing IBM SPSS software 21.0 and Python 3.13. Results are presented as mean values accompanied by their corresponding standard deviations (±SD). The choice of statistical tests was informed by assessment of data normality, evaluated using the Shapiro–Wilk test to determine Gaussian distribution. For hierarchical clustering and generation of heatmap visualizations, Ion Reporter software 5.20 (Thermo Fisher Scientific, USA) was employed. Categorical variables were analyzed using the Chi-square (χ^2^) test. Statistical significance was defined as a *p*-value less than 0.05.

## 3. Results

The subsections in this section are structured as follows: Section 3.1 through Section 3.3 are related to the analysis of data retrieved from PubMed publications, whereas Section 3.4 through Section 3.6 represent our own experimental data.

A total of 361 publications were found on the topic presented above. Genetic variability was discussed in 20 of them. A total of 26 studies were dedicated to the broad range of circulating autoantibodies (anti-SCL-70, Antitopoisomerase I, anti-RNA polymerase III, anti-centromere). The results of 12 studies revealed an association of active disease with an increase in specific cytokines in SSc patients’ serum (TGFβ, IL-1, IL-6, IL-33, IL-4, IL-13, IL-10, MCP-1, IFN-γ, TNF-α, IL-1α, IL-2, and CXCL10).

### 3.1. The Role of Autoantibodies in Systemic Sclerosis with Pulmonary Involvement

Autoantibodies represent a fundamental immunological feature of SSc, offering substantial diagnostic, prognostic, and pathogenetic insights. These biomarkers are instrumental in subclassifying the disease, forecasting internal organ involvement, and tailoring therapeutic strategies. The main biomarkers of interest for systemic sclerosis–interstitial lung disease (according to available literature) are given in Table 1 and Table 2.

The antibody profile in patients with SSc is crucial for predicting the current state of ILD: antibodies against topoisomerase I (ATA; also known as anti-Scl-70) [11,12] are primarily associated with the progression of pulmonary fibrosis, while antibodies against RNA polymerase III (anti-RNAP III) are linked to a higher risk of pulmonary hypertension and cardiac complications [3]. On the other hand, anti-centromere antibodies (ACA) are rarely detected in patients with SSc-ILD [7]. In 42% of patients [24], ILD complications were detected, which were linked to the presence of anti-RNAP-III and anti-Scl-70 antibodies. The presence of anti-Scl-70 antibodies was associated with a six-fold increased risk of developing ILD [24].

Anti-Scl-70 antibodies are closely linked to the diffuse cutaneous subset of SSc and are a strong predictor of early and progressive ILD. In contrast, ACA is generally associated with the limited form of SSc. Autoantibodies such as anti-Th/To (antibodies against a nucleolar 7-2/8-2 RNA-protein complex, specifically RNase MRP and RNase P), anti-Ro52 (antibodies directed against the Ro52/TRIM21 protein involved in ubiquitination and immune regulation), anti-U11/U12 RNP (antibodies targeting components of the minor spliceosome involved in RNA splicing), and anti-BICD2 (antibodies against the BICD2 protein, involved in intracellular transport) are increasingly recognized for their roles in pulmonary manifestations [25]. Jee et al. [26] recently proposed a composite biomarker index comprising SP-D (surfactant protein D), CA15-3 (cancer antigen 15-3), and ICAM-1 (intercellular adhesion molecule 1) that can effectively identify SSc-ILD.

In terms of vascular pathology, antibodies targeting endothelial and vascular receptors—including AECA (anti-endothelial cell antibodies), anti-AT1R (antibodies against angiotensin II type 1 receptor), and anti-ETaR (antibodies against endothelin-1 type A receptor)—have been associated with severe systemic and organ-specific vascular involvement. AECA have demonstrated correlation with extensive organ damage and impaired vascular integrity [27].

Overlap syndromes, combining features of SSc and myositis, frequently present with antibodies such as anti-U1-snRNP (antibodies against the U1 small nuclear ribonucleoprotein complex involved in mRNA splicing), and anti-PM/Scl (antibodies against a nucleolar exosome complex involved in ribosomal RNA processing). These serological markers help identify patients with mixed autoimmune phenotypes, guiding appropriate multidisciplinary care [28]. Additionally, non-organ-specific antibodies, including PIP4K2B (phosphatidylinositol-5-phosphate 4-kinase type 2 beta) and AKT3 (serine/threonine-protein kinase involved in cell survival and metabolism), have been detected in individuals with SSc. Although their exact functional role is yet to be fully defined, their presence may signify systemic immune dysregulation and enhanced fibrotic remodeling [25].

**Table 1 epidemiologia-06-00041-t001:** Antibodies in SSc-ILD: description and pathway features.

Markers	Description	Clinical Significance	Form of SSc	Reference
Anti-Scl-70	Anti-topoisomerase I antibody	ATA autoantibodies are strongly associated with the development of SSc-ILD, while ACA are protective for ILD.	diffuse	[29]
RNAP-III	Anti-RNA Polymerase III	Increased risk of PAH and skeletal muscle involvement, used to predict disease severity.	Diffuse	[30]
U1-snRNP	Anti-U1RNP antibodies	SSc patients with anti-U1RNP antibodies were more likely to develop ILD than SSc patients without anti-U1RNP. As a result, the hypothesis that these patients would experience a faster decline in FVC was raised.	limited	[31]
anti-BICD2	Intracellular protein bicaudal D2	For patients with SSc, a reduction in FEV1 and the carbon monoxide transfer coefficient has been observed.	-	[25,32]
anti-Th/To	Antibodies against Th/To ribonucleoprotein complex	Characterized by low organ damage, favorable ILD outcome, and good survival rates.	limited	[25]
anti-U11/U12 RNP	Antibodies against the U11/U12 minor spliceosomal ribonucleoprotein complex	Associated with ILD and fibrotic progression in SSc.	Overlap	[25]
CA15-3	Carbohydrate antigen 15-3 (MUC1)	Elevated levels indicate epithelial injury and fibrosis in SSc-ILD.	-	[26]
ICAM-1	Intercellular Adhesion Molecule 1	Indicator of endothelial activation and systemic inflammation.	-	[26]
Anti-C1q	Anti-C1q autoAbs	Anti-C1q autoAbs were frequently detected in patients with SSc, and their high levels predict the co-occurrence of pulmonary fibrosis or pulmonary arterial hypertension.	limited	[33]
KL-6	Krebs von den Lungen 6 glycoprotein	High KL-6 serum value (>923 U/mL) associated with more severe pulmonary functional impairment (large prospective trial) used to assess the severity of ILD.	-	[7,34]
SP-D	Surfactant protein D	Serum SP-D may be considered in several biomarkers for the severity of lung injury in SSc, including GER-associated lung injury.	-	[34]
Anti-U3-RNP	Anti-Fibrillarin	More prevalent in males with SSc with Afro-Caribbean ancestry rates between 7 and 11, who have a higher risk of developing pulmonary arterial hypertension (PAH).	Diffuse	[35,36]
Anti-PM/Scl	Anti-aminoacyl-tRNA synthetase	Anti-PM/Scl antibodies frequently resemble patients with Antisynthetase syndrome—an overlap myositis, interstitial lung disease and arthritis.	limited SSc, overlap	[27]
PIP4K2B	Autoantibodies against anti-phosphatidylinositol-5-phosphate 4-kinase type 2 beta (PIP4K2B)	Autoantibodies against PIP4K2B and AKT3 are increased risk of skin and lung fibrosis in patients with SSc.	-	[25]
AKT 3	AKT serine/threonine kinase 3 (AKT3)	Autoantibodies against PIP4K2B and AKT3 show increased risk of skin and lung fibrosis in patients with SSc.	-	[25]
anti-TRIM21/Ro52	anti-TRIM21/Ro52	The highest frequency of patients with lung fibrosis and PAH.	limited, diffuse, other autoimmune disease	[27]

### 3.2. Cytokines Profile in SSc-ILD

The association between serum cytokine levels and their correlation with lung involvement in SSc was investigated in several studies. Transforming growth factor-beta (TGF-β), interleukin-4 (IL-4), interleukin-8 (IL-8), C-C motif chemokine ligand 2 (CCL2), and C-C motif chemokine ligand 4 (CCL4) were found to have a strong correlation with lung involvement, while IL-6 and IL-7 showed lower levels of correlation [37,38]. In a subset of SSc-ILD patients, IL-6 also predicted mortality. IL-7 levels were predictive of a 20% or 30% decline in diffusing capacity for carbon monoxide (DLCO). The examination of these cytokines in serum suggests their potential as biomarkers for SSc, independent of those measured in bronchoalveolar lavage (BAL) [37]. De Lauretis et al. suggested that serum IL-6 levels may be a predictor of early disease progression in patients with mild ILD and could be used to tailor treatment [38].

Emerging evidence suggests that persistent fibrosis in SSc may be driven by endogenous Toll receptor 4 (TLR4) activators, such as tenascin-C, which is elevated in SSc tissues and promotes collagen production via TLR4 signaling. Mice deficient in tenascin-C exhibit reduced fibrosis, positioning it as a promising therapeutic target strategy for SSc-ILD [39,40].

Longitudinal studies also show that newly diagnosed SSc patients have elevated C-X-C motif chemokine ligand 10 (CXCL10) and C-C motif ligand 2 (CCL2) levels. CXCL10 is particularly associated with severe organ involvement, including the lungs and kidneys. Over time, CXCL10 levels decline while CCL2 levels remain stable, indicating a shift from a Th1-driven inflammatory phase to a Th2-dominant fibrotic phase [41,42]. Furthermore, baseline CXCL10 levels > 78.5 pg/mL are associated with a 2.74-fold increased risk of developing ILD, underscoring its prognostic value [43]. In pulmonary fibrosis, alveolar macrophages play a major role in the production of C-C motif chemokine ligand 18 (CCL18), which contributes to the development of the disease. In patients with SSc-ILD, elevated baseline serum levels of CCL18 are associated with a significant decline in lung function. This is defined as a decrease in forced vital capacity (FVC) by more than 10%. The risk of this decline is increased by a factor of 2.9 when CCL18 levels are elevated (HR = 2.90, *p* = 0.014) [34,44,45].

The transforming growth factor-β (TGF-β) plays a central role in the process of fibrosis and the dysregulation of the immune system towards inflammation. Injured or stressed cells produce TGF-β, which recruits and stimulates macrophages to secrete more TGF-β. This then upregulates genes responsible for extracellular matrix (ECM) production and leads to progressive fibrosis [46,47].

According to current literature, the main biomarkers of interest for systemic sclerosis-interstitial lung disease are given in Table 2.

**Table 2 epidemiologia-06-00041-t002:** Cytokines elevated in SSc-ILD.

Marker	Description	Clinical Significance	Reference
IL-7	Interleukin-7	Predictor for a decline of diffusion capacity (DLCO) by 20 or 30% in ILD patients.	[37]
IL-8	Interleukin-8	Serum IL-8 correlated with BAL IL-8 (r = 0.574, *p* = 0.006).	[37]
CCL4	C-C motif ligand 4	Significantly increased in patients with SSc presence of fibrosis, regardless of the subtype or stage of the disease and correlated with the severity of pulmonary fibrosis.	[48]
IL-6	Interleukin 6	Predictive marker of early disease progression in patients with mild ILD, and a marker for mortality in SSc-ILD patients.	[38,49,50,51,52]
Tenascin C	Tenascin C	Fibronectin-EDA and tenascin-C may activate Toll receptor 4 (TLR4) and lead to uncontrolled extracellular matrix (ECM) deposition in systemic sclerosis (SSc), which subsequently enhances transforming growth factor beta (TGF-β) signaling, thus promoting fibrotic responses.	[39,40]
TGF-β	Transforming Growth Factor-bet	TGF-β stimulates the expression of CUX1 isoforms, and the levels of ET-1, COL1, Wnt1, CTGF, and β-catenin increase after TGF-β treatment in normal and SSc lung fibroblasts.	[46,47]
CCL2 (MCP-1)	C-C motif ligand 2	A positive correlation has been found between CCL2 levels and the severity of pulmonary dysfunction, as measured by DLCO and FVC.	[41,42]
CXCL10	C-X-C motif chemokine ligand 10	CXCL10 is potentially involved in early SSc-ILD. Fibroblasts treated with serum or BAL fluid from patients with SSc overexpress CXCL10.	[43]
CCL18	C-C motif chemokine ligand 18	High CCL18 level is an independent predictor of pulmonary function decrease in SSc-ILD (>10% in forced vital capacity).	[33,44,45]

### 3.3. Genetic Predisposition of Lung Injury in Systemic Sclerosis in Different Nations

Genetic predisposition to SSc is supported by epidemiological data showing the relative risk of SSc. For instance, SSc occurs significantly more often in families with a history of SSc (1.6%) than in the general population (0.026%) [53].

A recent study of serum samples from a group of patients with SSc from Paris, France, and Oslo, Norway, found that surfactant protein D (SP-D) might be a potential diagnostic biomarker for ILD associated with SSc, while Krebs von den Lungen 6 glycoprotein (KL-6) can be used to assess the severity of pulmonary fibrosis. CCL18 also showed potential as a predictor for the progression of ILD in SSc patients [33].

Chevalier et al. [30] showed that SSc patients with anti-U1-snRNP antibodies were more likely to develop ILD than SSc patients without anti-U1-snRNP. As a result, the hypothesis that these patients would experience a faster decline in forced vital capacity (FVC) was raised. However, recent data of the SSc-ILD patients with anti-U1-snRNP antibodies have worse baseline lung function, with a lower mean forced vital capacity (82.0% vs. 86.0%, *p* < 0.001) and diffusing capacity for carbon monoxide (57.0% vs. 60.5%, *p* = 0.003). They also have a higher prevalence of limited cutaneous systemic sclerosis, joint synovitis, and myositis. Despite these differences, their trajectories of FVC decline and mortality over three years are comparable to those without the antibodies [54]. This may improve risk stratification and facilitate more personalized approaches in clinical management.

Risk factors for SSc-ILD include male gender, diffuse cutaneous SSc, African-American ancestry, and the presence of anti-Scl-70 (anti-topoisomerase I) antibodies [48]. The study, which involved a large sample of the U.S. population, revealed that individuals of African ancestry had a higher prevalence ratio, after adjusting for other factors, compared to individuals of European ancestry [55], while Choctaw Native Americans exhibited the highest reported prevalence—66 cases per 100,000 population [56]. SSc patients of European descent tend to have a lower frequency of ILD, a slower rate of lung function decline, and improved survival outcomes compared to those of African, Japanese, and Choctaw ancestry [57].

Over the past decade, researchers have identified an increasing number of genetic markers linked to SSc-ILD. This has highlighted the importance of genetic factors in the development of this condition. The Signal Transducer and Activator of Transcription 4 (STAT4) gene codes for the STAT-4 protein, which regulates cytokine signaling from interleukin-12, interleukin-23, and interferon-gamma in T cells and monocytes [58]. Studies, including research conducted on the Russian population, have shown that the polymorphism STAT4 (rs7574865 (G/T)) is significantly involved in susceptibility to SSc and its associated clinical manifestations, such as ILD [59].

A search performed in the PubMed database for studies published in the last 10 years related to the SSc-ILD association genes revealed interferon regulatory factor 5 (IRF-5), signal transducer and activator of transcription 4 (STAT4), DNAX accessory molecule 1 (CD226), and interleukin-1 receptor-associated kinase 1 (IRAK-1) (Table 3).

Wenjie Zhao1 et al. [60] categorized the susceptibility genes for systemic sclerosis (SSc) according to their biological function into three main groups: adaptive immunity, innate immunity, and non-immune genes. Among the genes analyzed, four out of seven were found to be significantly associated with the presence of pulmonary fibrosis (PF) in patients with SSc. These include IRF5 rs2004640 (T allele)—OR 1.12 (95% CI: 1.02–1.22), *p* = 0.0139; STAT4 rs7574865 (T allele)—OR 1.15 (95% CI: 1.07–1.47), *p* = 0.0053; CTGF G-945C (C allele)—OR 1.42 (95% CI: 1.18–1.71), *p* = 0.0002; IRAK1 rs1059702 (T allele)—OR 1.20 (95% CI: 1.05–1.37), *p* = 0.007. These results suggest a potential role of both innate immune and fibrotic regulatory pathways in the pathogenesis of PF in SSc patients. Two genes, CD247 and CD226, are known to be involved in T-cell activation and function, but their specific role in the development of pulmonary fibrosis remains unclear [60].

Another study identified pathways of SSc-ILD highlighting matrix metalloproteinase 7 (MMP7) as a gene that is consistently upregulated and potentially involved in fibrosis through G-protein coupled receptor signaling. The enrichment of immune and inflammatory pathways, including the tumor necrosis factor (TNF) signaling pathway, suggests that immune dysregulation plays a significant role in SSc-ILD pathogenesis [61].

**Table 3 epidemiologia-06-00041-t003:** Genetic predisposition of lung injury in systemic sclerosis in different nations.

Genes	Polymorphism	Function	Population	Reference
IRF5	rs10488631, rs12537284, rs4728142	Associated with lower IRF5 transcript levels, was predictive of longer survival and milder ILD in patients with SSc.	Caucasian	[58]
CD247	rs2056626	Associated with T cell activation and function.	Caucasian Turkish Chinese	[58,60,62]
CD226	rs763361	CD226 (also known as T cell Ig and ITIM domain) are involved in the T-cell activation and function.	Iranian	[62]
IRF 5	rs2004640	Important regulatory factors in controlling innate immune responses.	French	[60]
IRF5	rs4728142	Lower IRF5 transcript levels are associated with longer survival and milder ILD.	Caucasian Turkish	[60]
STAT4	rs3821236	Important in predisposition to SSc-ILD.	Caucasian	[58]
STAT4	rs7574865	Important regulatory factors in controlling innate immune responses and participating in the extracellular matrix synthesis in the pathogenesis of in SSc-ILD.	Russian	[58,59,63]
IRAK1	rs1059702	Mediator of TLR/IL-1R signaling; contributes to proinflammatory and profibrotic responses in SSc-ILD.	European Caucasian	[60]
SPINT2	-	Serine protease inhibitor; suppresses epithelial-to-mesenchymal transition and fibrosis. Overexpressed in activated myofibroblasts.	Not specified	[64]
MFAP5	-	ECM protein upregulated in myofibroblasts; promotes collagen deposition and tissue remodeling in lung fibrosis.	Not specified	[64]
CDKN2C (p18)	CDK inhibitors	The potential role of the cdkn2c (p18), associated with protein expression, is increased in progressing SSc-ILD lung sections.	Not specified	[20]
PTGS2 COX-2	-	Inflammatory mediator gene downregulated via ncRNAs; positively associated with the occurrence of SSc-ILD and abnormal immune cell infiltration. Potential factor for the progression of SSc-ILD to malignancy.	Nor specified	[65]
CTLA-4	318C/T	Immune checkpoint gene; polymorphism associated with increased susceptibility to SSc.	Not specified	[66]
CD38	-	Markers of plasma cells, co-expressed in fibrotic SSc lungs, indicating B cell and antibody-producing cell involvement.	Not specified	[67]
CD138	-	Markers of plasma cells; co-expressed in fibrotic SSc lungs, indicating B cell and antibody-producing cell involvement.	Not specified	[67]
miR-155	-	miRNAs regulate gene expression; expression levels correlate with lung function and fibrosis progression in SSc-ILD.	Not specified	[68]
miR-143	-	miRNAs regulate gene expression; expression levels correlate with lung function and fibrosis progression in SSc-ILD.	Not specified	[68]
MMP7	-	Matrix metalloproteinase involved in ECM degradation and fibrosis; may modulate GPCR signaling and promote fibrotic remodeling.	Not specified	[61]

IRF5 = Interferon Regulatory Factor; CD247 = CD3 zeta chain; CD226 = DNAX accessory molecule 1; STAT4 = Signal Transducer and Activator of Transcription4; IRAK1 = Interleukin-1receptor Associated Kinase 1; SPINT2 = Serine Peptidase Inhibitor type 2; MFAP5 = Microfibrillar-Associated Protein 5; CDKN2C (p18) = Cyclin-Dependent Kinase Inhibitor 2c; PTGS2 (COX-2) = Prostoglandin-Endoperoxide Synthase 2; CTLA-4 = Cytotoxic T-Lymphocyte-Associated Protein 4; CD38 = Cluster Of Differentiation 38; CD138 = Syndecan-1; miR-155 = MicroRNA-155; miR-143 = MicroRNA-143.

### 3.4. Immunogenetic Profiling of Kazakh Patients with SSc-ILD

The 30 Kazakh patients with SSc were initially examined in the present study, apart from which 14 patients (47%) had lung involvement such as SSc-ILD and arterial hypertension (Table 4). The clinical and genetic data for these 14 patients only were included in the further analysis, i.e., the percentages given below relate to the subgroup of these 14 patients.

Rodnan skin score was equal 15.93 ± 7.04 and EScSG—4.27 ± 2.09. The clinical manifestations for all these patients with lung involvement (n = 14) are given in Table 4, all of whom also presented with skin lesions. Among them, four patients had both pulmonary arterial hypertension (PAH) and interstitial lung disease (ILD). Lung impairment manifested as SSc-ILD, with features of either non-specific interstitial pneumonia (NSIP, more common) or usual interstitial pneumonia (UIP) observed on high-resolution computed tomography (Figure 1). A total of five patients (36%) were diagnosed with secondary Sjögren’s syndrome, and cardiovascular disorders were present in two patients (14.3%). Joint involvement, including arthralgia and arthritis, was noted, along with vascular manifestations such as Raynaud’s phenomenon, which was observed in 12 out of 14 patients (86%). Additionally, 10 out of the 14 patients (71.4%) with lung involvement also had frequent gastrointestinal tract lesions.

The mean age of the disease duration was 9.0 ± 6.8 years. According to the prescribed treatment, eight patients obtained steroids with the average dose of 7.17 ± 3.28 mg of methylprednisolone. Other medications included Mycophenolate mofetil (one patient), in combination with Methotrexate (one patient), or with Leflunomide (one patient). D-Penicillamine was given to two patients, while Leflunomide was prescribed to two patients with prominent polyarticular syndrome, with Mycophenolate mofetil (one patient), or with D-penicillamine (one patient). Antifibrotic drug Nintedanib in combination with Leflunomide was prescribed to one patient. One patient did not receive any medications.

### 3.5. Autoantibodies and Their Association in the Development of SSc-ILD

Antinuclear factor showed a positive titer (>1:160) in 12 patients and a negative titer (<1:80) in 2 patients (reference interval < 1:80 negative, ≥1:80 positive). The main types of Hep-2 cell glow observed in the experimental group of SSc patients were nuclear speckled (AC-4/5) and centromere (AC-3). Various types of the cytoplasmic glow (AC-19, AC-20, AC-21) were also detected (Table 5).

The mean titer of antinuclear antibodies was equal to 1200 ± 1446, with the most common antibodies to CENP-B, Sm, SS-A/60, U1-snRNP, RNP/Sm, rib-P0, SS-B, SS-A/60, SS-A/52, RNP/Sm, ribP0, U1-snRNP, ds-DNA, and Nucleosome at concentrations above 10.0 IU/mL (reference interval: <8.0 IU/mL negative, 8.0–10.0 IU/mL borderline, >10.0 IU/mL positive). In the control group, the levels of ANA < 1:80 and antibodies were negative.

There have been several prognostic biomarkers and potential treatment targets identified that could help predict and manage the progression of lung fibrosis in patients with SSc-ILD. Several serological markers, including interleukin-6 (IL-6), C-C motif ligand 18 (CCL-18), CXCL10, and various other inflammatory cytokines and chemokines, have been proposed; however, they require further investigation in a targeted patient population [43].

Kazakh patients with SSc-ILD demonstrated (see the Figure 2) the presence of typical SSc antibodies: CENP-B (antibodies against centromere protein B), SS-A/52 (antibodies against the 52 kDa Ro/SSA protein, involved in immune regulation), Sm (antibodies against Smith antigen, a component of the spliceosome), U1-snRNP (antibodies against U1 small nuclear ribonucleoprotein, important in RNA splicing), RNP/Sm (antibodies targeting both ribonucleoproteins and Smith antigens), Ribosomal P0 (antibodies against the ribosomal P0 phosphoprotein, related to RNA translation), SS-A/60 (antibodies against the 60 kDa Ro/SSA protein), SS-B (antibodies against La/SSB, involved in RNA metabolism), dsDNA (antibodies against double-stranded DNA), Nucleosome (antibodies targeting DNA-histone complexes), and Scl-70 (antibodies against DNA topoisomerase I, highly specific for diffuse SSc). Additionally, some of these antibodies—such as CENP-B, Sm, Ribosomal P0, Nucleosome, and dsDNA—are commonly associated with other autoimmune diseases, highlighting the serological overlap often observed in connective tissue disorders.

The level of IL-6 remains within the reference interval in 10 patients (reference interval 0.0–10.0 pg/mL). Four patients (28.6%) had elevated IL-6 levels (>10.0 pg/mL), with the following clinical features: Skin involvement (dense edema, hyperpigmentation, or depigmentation) and joint impairment (arthralgia or polyarthritis) were observed in all four patients (28.6%) with increased IL-6 levels. Pulmonary arterial hypertension (PAH) was present in two of the four patients (14.3%) with elevated IL-6, including a patient with an IL-6 level of 21.7 pg/mL and ESR of 64 mm/h. Esophagitis was diagnosed in three of the four patients (21.4%) with increased IL-6. The patient with the highest IL-6 level (26.3 pg/mL) presented with severe skin sclerosis, ILD, reflux esophagitis, and cytoplasmic antibody patterns (AC-19/20). Detailed information is provided in Appendix A. In the control group (n = 18), IL-6 levels remained within the reference range in all individuals.

### 3.6. Genetic Variability

Genetic analysis of samples of the cohort of SSc-ILD revealed multiple likely pathogenic variants in a panel of 120 genes. These include Sterile Alpha Motif Domain Containing 9 Like (SAMD9L), as well as the following genes: Lymphocyte Antigen 96 (Ly96), REL Proto-oncogene, Nuclear Factor kappa B (NF-κB) subunit (REL), Interleukin-1 receptor-associated kinase-1 (IRAK1), Recombination signal binding protein for immunoglobulin kappa J region (RBPJ), Interleukin 6 signal transducer (IL6ST), Integrin subunit alpha 2b (ITGA2B), ATP binding cassette subfamily C member 2 (ABCC2), Interleukin 6 receptor (IL6R), and IKAROS Family Zinc finger 3 (IKZF3) (Figure 3). For the SAMD9L gene, three distinct variants—chr7:92762447 CA/C, chr7:92764981 T/TT, and chr7:92761606 GT/G—were identified in nine patients, representing 64.3% of the screened subgroup. According to guidelines of the American College of Medical Genetics and Genomics (ACMG) classification, all three variants were considered likely pathogenic.

Previous studies have shown that SAMD9L mutations are associated with cytopenia and inflammatory disorders [69]. In our cohort, all nine patients carrying these variants also demonstrated pronounced autoimmune features. Recent studies have linked SAMD9L mutations to severe autoinflammatory syndromes, characterized by ILD, elevated inflammatory markers, cytopenias, and immune dysregulation, which closely resemble clinical and immunological features seen in SSc-ILD [70].

In our cohort, patients with likely pathogenetic variants of SAMD9L, particularly those in a compound heterozygous state, showed a combination of pulmonary arterial hypertension and ILD, suggesting that SAMD9L dysfunction may contribute to vascular damage, fibrosis, and chronic inflammation, all of which are central to SSc-ILD pathogenesis. Detailed information on these variants is provided in Appendix A.

A likely pathogenic variant of the REL gene (chr2: 61149099 GT/G) was identified in one patient from the main group (n = 14). The REL gene encodes for c-Rel, which is a member of the NF-κB family of transcription factors. This protein is predominantly expressed in B and T lymphocytes and regulates genes involved in lymphocyte proliferation. Alterations in the REL gene have been linked to autoimmune conditions such as rheumatoid arthritis, psoriasis, celiac disease, and certain hematological malignancies [71]. These findings support the potential role of dysregulation of the REL gene in immune-mediated diseases, including systemic sclerosis.

Two distinct variants were found in IRAK1 gene (chrX:153278833 GCCCG/GCC and GCC/GCCG), observed in 4 out of 14 patients (28.6%), making IRAK1 one of the more frequently mutated genes in our cohort. Our patients carrying these IRAK1 changes tended to have more severe clinical features, aligning with earlier studies [59] linking IRAK1 variants to diffuse cutaneous forms and interstitial lung involvement. Variants in miR-146a may influence disease susceptibility by affecting the regulation of target proteins such as IRAK1, IRAK2, and TRAF6, which play key roles in TNF and NF-κB signaling pathways, thereby modulating inflammatory responses [72].

One variant of LY96 (chr8:74922341 CT/C) gene was detected. About 57.1% of carriers exhibited pronounced inflammatory profiles according to ESR level; however, IL-6 was within normal range. Possibly reflecting the role of Ly96 (also known as MD-2) modulates immune pathways in rheumatic diseases and as a co-receptor in Toll-like receptor signaling [73].

A heterozygous likely pathogenic variant in ITGA2B (chr17:42453072 G/GC, ref GCC) was identified in one of the 14 patients (7.1%). Interestingly, we found that the carrier showed significant platelet activation, along with usual fibrosis markers. However, due to the small sample size, it was unclear whether ITGA2 B changes significantly contributed to the severity of ILD development or vascular complications.

A single variant in the AIRE gene was found in our study group (7.1%). The affected individual had a mixed autoimmune condition, including autoimmune thyroiditis (Table 4). Previous studies on genome-wide associations in Asian populations (Japanese and Chinese) linked AIRE polymorphisms to an increased risk of developing rheumatoid arthritis [73,74]. Given that AIRE has central role in immune tolerance, even heterozygous variations may contribute to more widespread autoimmune dysregulation. However, this needs to be confirmed through studies involving larger groups of patients.

Moreover, one novel variant was found in the TREX1 (chr3:48508185 T/TC) which was likely pathogenic. This variant was observed in a single patient in our main group. Experimental studies have shown that TREX1 dysfunction leads to the accumulation of dsDNA, chronic immune activation, the production of autoantibodies, and the deposition of immune complexes—hallmarks of systemic autoimmune disease [75]. Our findings suggest that impaired DNA clearance by TREX1 contributes to inflammation mediated by interferon and supports its role in the pathogenesis of connective tissue diseases, such as SSc-ILD.

In our study, a pathogenic variant in the CD40 gene was identified in three individuals (21.4%) from a subgroup of 14 patients with SSc-ILD. This variant was located at chr20:44750489 and was found to be GA/G. The CD40 molecule is known to play a central role in the co-stimulation of T and B cells and has also been linked to the pathogenesis of autoimmune diseases. A previous meta-analysis found a significant association between the CD40 rs4810485 polymorphism and susceptibility to rheumatoid arthritis and systemic lupus erythematosus in European populations [76]. However, the variant we identified in our study is different from rs4810485. Nevertheless, our findings support the idea that CD40 dysregulation may be involved in immune-mediated diseases like systemic sclerosis. Further studies are needed to understand the functional consequences of this particular variant and its role in disease development.

Two patients (14.3%) from the subgroup were found to carry a pathogenic variant in the SMAD family member 3 (SMAD3) gene (chr15:67477130 AT/A). This variant was not registered in the control group. It is worth noting that the two patients with SSc-ILD, with altered variants in the SMAD3 gene, have a vascular component in the clinical picture: vascular damage, Raynaud’s syndrome. These findings may suggest a potential role of SMAD3 in the vascular manifestations of SSc.

One variant occurred in the genes AFF3, IKZF3, TREX1, IL18, PRKCQ, and PXK. The variant is classified as a variant of uncertain significance.

Finally, variants in the IL6R gene (chr1:154378136 GC/G) were detected in three patients (21.4%) from the study group; these variants were not found in the control group. The chr1:154378136 GC/G variant is classified as likely pathogenic, and the chr1:154401686 G/A variant is a variant of uncertain significance. These findings may indicate a potential involvement of IL6R pathway alterations in the immunopathogenesis of SSc-ILD, although further studies are needed to confirm their clinical relevance.

Overall, although many of these variants occurred at relatively low frequencies, the overall picture highlights the complex polygenic nature of SSc-ILD. Multiple immune-signaling genes, particularly SAMD9L, REL, ITGA2B, AIRE, TREX1, CD40, IRAK1, TNFAIP3, and Ly96, may act together to influence susceptibility to the disease and the course of the disease.

## 4. Discussion

Pulmonary involvement in systemic sclerosis remains a leading contributor to morbidity and mortality, reflecting the multifaceted interplay between immune abnormalities, fibrotic mechanisms, and underlying genetic susceptibilities.

Immunogenetic analysis plays a significant role not only in the early verification of the diagnosis, but also in the selection of appropriate therapy and the prevention of patient disability. Based on the systematic review results, we created a preliminary list of genes and autoantibodies associated with SSc-ILD. Depending on the presence of certain antibodies and genes variations, the treatment of systemic sclerosis with lung involvement may begin immediately with the use of antifibrotic therapy.

The underlying mechanism of SSc-ILD pathogenesis is thought to begin with a persistent injury to lung cells, which triggers profibrotic stimuli (Figure 4). Endothelial cells play a significant role in the pathophysiology of SSc. Their overactivation/aggressive release leads to cell activation and the launch of an inflammatory response, which can result in chronic inflammation and the development of ILD. ET-1 is known for vasoconstrictive properties, playing a role in the differentiation of myofibroblasts and the production/deposition of extracellular matrix (ECM). In patients with SSc-ILD, the levels of ET-1 are increased in BAL and serum [77]. The histological studies shown that patients with SSc-ILD have an overexpressed level of ET-1 in lungs [78].

SSc-ILD patients with anti-U1RNP antibodies have worse baseline lung function, with a lower mean forced vital capacity (82.0% vs. 86.0%, *p* < 0.001) and diffusing capacity for carbon monoxide (57.0% vs. 60.5%, *p* = 0.003). They also have a higher prevalence of limited cutaneous systemic sclerosis, joint synovitis, and myositis. Despite these differences, their trajectories of FVC decline and mortality over three years are comparable to those without the antibodies.

Studies have confirmed that IL-6, a pro-inflammatory cytokine, is involved in the pathogenesis of SSc [38]. Previous studies have found that serum IL-6 is a predictive marker for the early decline in pulmonary function and mortality in patients with SSc-ILD [38]. Interleukin 6 (IL-6) is involved in the development of pulmonary fibrosis by stimulating the expression of genes related to collagen, promoting the growth and differentiation of fibroblast and myofibroblast cells, inhibiting the apoptosis of T cells, and regulating the balance between Th17 cells [79,80]. In an animal model of SSc, knocking out the IL-6 gene significantly reduced lung inflammation and collagen deposition in mice.

SSc-ILD shares key pathogenic pathways and exhibits distinct features, such as an increased presence of senescent and apoptosis-resistant myofibroblasts. The p53 signaling pathway is the most prominent molecular signature in SSc-ILD, closely linked to impaired gas exchange, cellular senescence, and resistance to apoptosis. Essential regulatory genes identified include eukaryotic translation elongation factor 2 (EEF2), eukaryotic elongation factor 2 kinase (EEF2K), phosphorylase kinase catalytic subunit gamma 2 (PHKG2), vascular cell adhesion molecule 1 (VCAM1), protein kinase cAMP-activated catalytic subunit beta (PRKACB), integrin subunit alpha 4 (ITGA4), cyclin dependent kinase 1 (CDK1), cyclin dependent kinase 2 (CDK2), fibronectin 1 (FN1), and histone deacetylase 1 (HDAC1) [81].

Transforming growth factor-β is a key mediator that triggers the differentiation of myofibroblasts, stimulates the production of ECM, and inhibits the production of metalloproteinases (MMP) [81]. TGF-β induces the expression of the CUX1 isoforms p200, p150, p110, p75, p30, and p28 in SSc lung fibroblasts [46]. CUX1 isoforms act as transcriptional activators, increasing COL1 and pro-fibrotic cytokines (CTGF, Wnt1, ET-1, β-catenin). CUX1 binds to the IS4 (−19.51 kb) domain of COL1A2 enhancer region and stimulates fibroblast-to-myofibroblast transition (α-SMA expression, production of extracellular matrix molecules (EMM) such as COL1). TGF-β is produced in macrophages, epithelial cells, and activated myofibroblasts, which perpetuates the abnormal wound healing process by inducing the production of extracellular matrix, promoting the recruitment and activation of fibroblasts and transition into myofibroblasts [82].

The peptide involved in TGF-β-induced fibrosis is connective tissue growth factor (CTGF), which promotes the recruitment of myofibroblasts to produce collagen 1 and fibronectin, leading to the overproduction and accumulation of extracellular matrix [83]. Serum levels and tissue expression of matrix metalloproteinase-12 (MMP-12) were elevated in patients with SSc-ILD, correlated with the severity of lung restriction, and are associated with the severity of skin and peripheral vascular damage [83]. A higher level of CTGF has been identified in SSc-ILD patients and was correlated with the degree of skin sclerosis and severity of pulmonary fibrosis [84]. Moreover, there is a positive correlation between the severity of SSc and the serum levels of CTGF, and CTGF polymorphisms were associated with lung involvement [59,85]. Prolonged overexpression of these cytokines leads to chronic stimulation of fibroblasts (fibrosing). The abilities of myofibroblasts lie in the activation and enhancement production of CTGF, TGFβ, and ECM, which stimulate the formation and collagen type I production. TGFβ induces the expression of ET-1, CTGF, and VEGF in myofibroblasts.

Myofibroblasts produce various cytokines and chemokines to aid in the recruitment and facilitate the function of (innate) immune cells: IL-1, IL-6, IL-8, and monocyte chemoattractive protein 1 (MCP-1) [86]. This process is likely to be intensified by persistent inflammation and the interaction between inflammatory activity and fibrosis, with IL-6 emerging as a key mediator [87]. The formation of myofibroblasts is increased against the background of weakening of the intrinsic pathway of apoptosis in them [87]. The intrinsic pathway is initiated by the release of cytochrome C from mitochondria, which subsequently forms apoptosomes. These cellular structures activate caspase-dependent pathways to initiate apoptosis. However, in SSc, the balance between pro- and anti-apoptotic proteins is disrupted, leading to an increase in the activity of pro-apoptotic pore-forming proteins, such as BAX, and their inhibitors, such as BCL2 and BCL2-XL [88]. The presence of pathological myofibroblasts significantly negatively affects lung function (Figure 4). Their matrix-producing ability destroys alveolar architecture and increases interstitial space thickness, which both hamper respiration. Furthermore, the presence of myofibroblasts can induce stenosis—the abnormal narrowing of blood vessels—and blood vessel narrowing is further enhanced by myofibroblasts’ expression of ET-1, a potent vasoconstrictor. This hampers pulmonary blood flow and consequently induces strain on the right heart ventricle.

The analysis of currently available up-to-date data revealed the association of IRF-5, STAT4, DNAX, accessory molecule 1 (CD226), and IRAK-1 genes with SSc-ILD. In our study, we identified pathogenic or likely pathogenic variants in LY96, IRAK1, TREX1, CD40, REL, SAMD9L, SMAD3, and ITGA2B. Given these findings, conducting further genetic studies in the Kazakh population is essential to identify population-specific genetic markers associated with SSc-ILD susceptibility and disease severity.

In the majority of cases, late treatment can lead to faster progression of the ILD. Identifying effective biomarkers can prevent the progression of fibrosis. For instance, anti-Scl-70 antibodies are suggested as a predictor for faster decline in FVC in patients with SSc-related ILD [89]. In our research, increased level of Anti-Scl-70 was detected in one male patient, with a rapidly progressive course of SSc-ILD with pulmonary arterial hypertension, Raynaud’s syndrome, skin injury, esophagitis, and arthritis. The same patient suffered from secondary Sjogren’s syndrome with anti-SS-A/60 and SS-A/52 antibodies.

It should be noted that our study has several limitations. The key limitation is that our sample was relatively low (n = 14 patients with SSc-ILD) to represent accurately the Kazakh population. We aim to acquire more data in further studies. Another principal limitation which is directly linked to the first one is that we cannot accurately stratify patients according to general population-related features like sex and age. Our present results rather demonstrate that the new genetic variants of SMAD3, REL, LY96, IRAK1, TREX1, CD40, REL, SAMD9L and ITGA2B in SSc-ILD do exist in Kazakh population and that further research is needed to reveal if the alterations are unique for Kazakh nation.

In conclusion, various parts of the immune system become active in SSc-ILD, leading to the identification of many potential biomarkers and therapeutic targets. The trend is towards the use of biomarker panels in combination with complex multifactor analysis, machine learning, and artificial intelligence to monitor disease activity and evaluate the effectiveness of treatment.

Overall, our study highlights the immunogenic complexity of systemic sclerosis-associated interstitial lung disease, particularly within the Kazakh population, offering new insight into population-specific risk factors. Our findings contribute to a more refined understanding of disease variability and progression. These results pave the way for precision medicine approaches that integrate genetic and immunological data to enhance early detection, forecast disease trajectory, and guide individualized treatment. Further research including whole-genome sequencing, expanding the cohort of patients, and transcriptomic studies are vital to advance our knowledge of SSc-ILD and to translate molecular discoveries into meaningful clinical outcomes.

## Figures and Tables

**Figure 1 epidemiologia-06-00041-f001:**
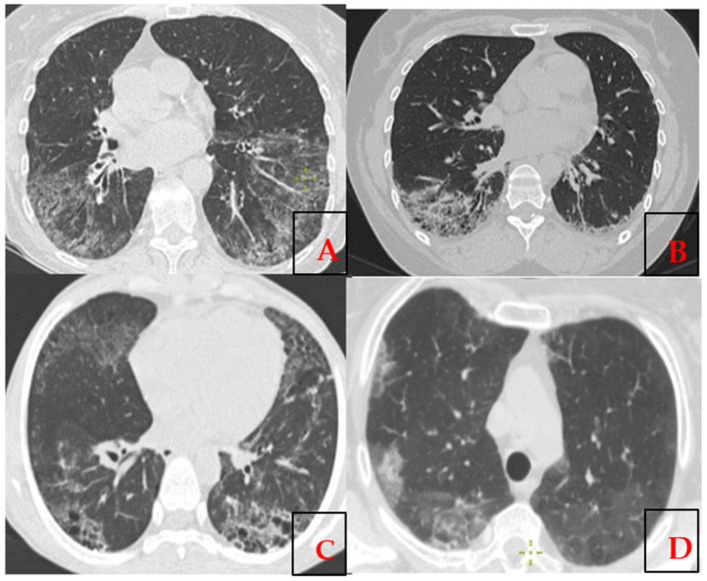
The high-resolution computed tomography axial projection of the chest demonstrates (**A**,**B**) features of usual interstitial pneumonia (UIP); (**C**,**D**) signs of non-specific interstitial pneumonia (NSIP).

**Figure 2 epidemiologia-06-00041-f002:**
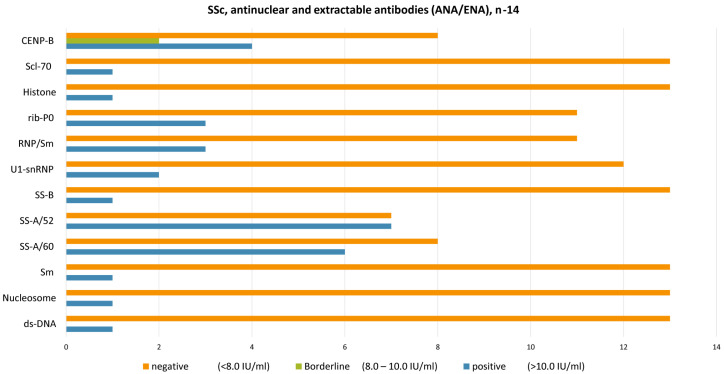
A graphic representation of the spectrum of antibodies, ANA/ENA.

**Figure 3 epidemiologia-06-00041-f003:**
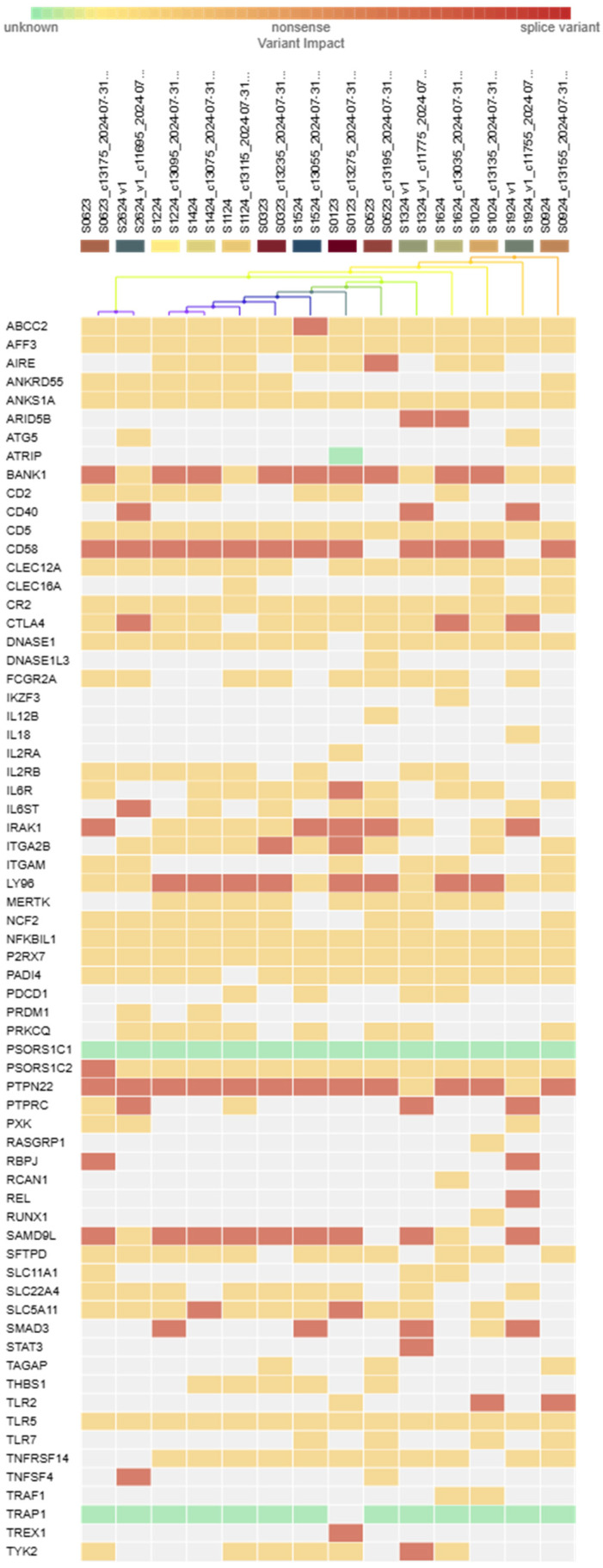
The heatmap illustrates the distribution of germline mutations across 67 target genes (in a panel of 120 genes) in 14 samples of systemic sclerosis interstitial lung disease (SSc-ILD). The rows represent only those targeted genes which were found altered in variants (n = 67), while the columns represent individual samples. The samples are grouped based on their mutation status and predicted impact, as determined by the IonReporter software (Thermo Fisher Scientific, version 5.2). The color coding reflects the variant classification: Green: variant with unknown impact, Orange: missense variant, Red: nonsense variant, Maroon: splice variant, White: no detected variant (above the threshold). The dendrograms of hierarchical clustering are displayed at the top of each heatmap, illustrating clusters of samples with similar genetic mutation patterns. To be included in the analysis, a minimum of 30 reads were required. The genes on the *y*-axis are arranged in alphabetical order. Variant Frequency Analysis: To assess potential differences in the distribution of genetic variants between groups, the Chi-square (χ^2^) test were employed. None of the loci analyzed showed statistically significant variation at the standard threshold of *p* < 0.05.

**Figure 4 epidemiologia-06-00041-f004:**
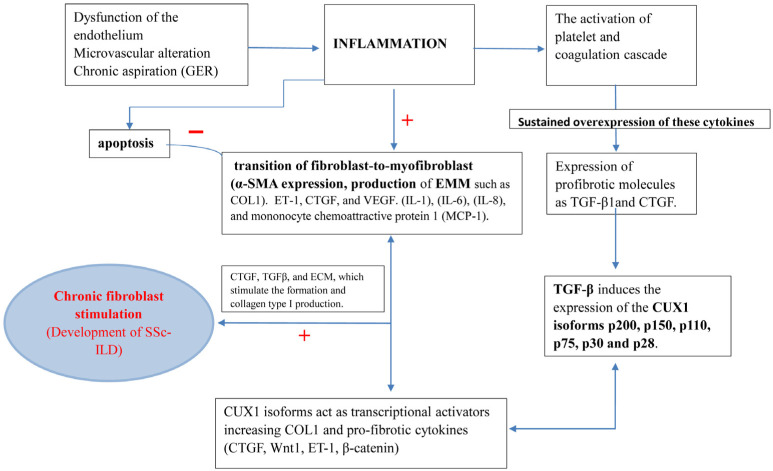
Pathogenesis of the development of interstitial lung disease in patients with SSc (“+” Stimulation; “−” Inhibition).

**Table 4 epidemiologia-06-00041-t004:** Clinical manifestations in patients with SSc-ILD (n = 14).

Clinical Manifestation	Number of Patients (n = 14)
Lung involvement:	14 (100%)
PAH + ILD	4 (28.6%)
ILD	13 (92.9%)
Skin involvement	14 (100%)
Joint involvement:	13 (92.9%)
Sclerodactyly—2 out of 13 (15%)	2 (14.3%)
Joint functional impairment (Grade II)	6 (42.9%)
Joint functional impairment (Grade I)	1 (7.1%)
Gastrointestinal (GI) tract involvement	10 (71.4%)
Raynaud’s phenomenon	12 (85.7%)
Cardiosclerosis	2 (14.3%)
Sjögren’s syndrome	5 (35.7%)

**Table 5 epidemiologia-06-00041-t005:** The results of ANA and ENA tests in patients with SSc-ILD.

SSc, Antinuclear and Extractable Antibodies (ANA/ENA), *n* = 14	ds-DNA	Nucleosome	Sm	SS-A/60	SS-A/52	SS-B	U1-snRNP	RNP/Sm	rib-P0	Histone	Scl-70	CENP-B
Positive (>10.0 IU/mL)	1	1	1	6	7	1	2	3	3	1	1	4
Borderline (8.0–10.0 IU/mL)	0	0	0	0	0	0	0	0	0	0	0	2
Negative (<8.0 IU/mL)	13	13	13	8	7	13	12	11	11	13	13	8

## Data Availability

The research data were not published in publicly available repositories because all relevant data are included to the manuscript. The raw data related to the genetic analysis (immunoblotting, sequencing, etc.) are available from the corresponding author by request.

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
