# Peer review of "Systemic Sclerosis with Interstitial Lung Disease: Identification of Novel Immunogenetic Markers and Ethnic Specificity in Kazakh Patients"

_epidemiologia, 2025, doi:10.3390/epidemiologia6030041_

Round 1
Reviewer 1 Report
Comments and Suggestions for Authors
I remember to the authors that when in the text for the first time is introduced an abbreviation is necessary to put a description and after the abbreviation between (). Please check the articles. I correct only some. The authors introduced descriptions of abbreviations and sometimes did not in the text. Alternatively introduce description of all abbreviations in line 619. Sometimes , data or tables are not clear.
Line 199 to 210: Please introduce description of abbreviation in the text, for example anti Th/To: antibodies against a nucleolar 7-2/8-2 RNA-protein complex, specifically two endoribonucleases: RNase MRP and RNase P. Anti AT1R: antibodies that bind to the angiotensin II type 1 receptor (AT1R), and so on.
Line 208 and 210: anti-PM/ScI, PIPk2B and AKT3. Please introduce description of abbreviation: antibodies against …
Line 201: introduce reference about Jee et al. and introduce space between point and Jee
Line 210: Introduce space between including and PIP4K2B.
Line 216: introduce transforming growth factor bet (TGF-b), Interleukin 4 (IL-4), ….
Line 233: introduce space between point and Furthermore
Table 2: correct the last line. Some antibodies are cited in 3.1 but not introduced in Table 2 Why? Have they clinical significance or not? (Ti/To, anti Ro52, anti BICD2, Sp-D, CA15-3, ICAM 1 etc)
Line 233: introduce space between point and furthermore.
Line 257 to 260: introduce before abbreviation a description of antibodies as in line 199-210 and 208-210. Example: CENP-B (antibodies against centromere) and so on.
Line 251 to 256: On my opinion Is better to put the phase in “cytokines profile in SSc-ILD” and begin the 3.3 directly by “Genetic predisposition…””?
Line 257 to 264: cancel. Results and description also shown in 3.5
Line 270 to 273: introduce Surfactant protein D (SP-D), protein Krebs von den Lungen-6 (KL-6).
Line 276: introduce Forced Vital Capacity (FVC)
Line 295: introduce Signal transducer and activator of transcription 4 (STAT 4)
Line 301: as in line 295: introduce Interferon regulatory factor 5 (IRF5), IRAK-1….
Line 314: introduce Serum matrix metalloproteinase 7 (MMP7)
Table 3. introduce description of abbreviations that are not present in chapter 3.3
Line 349: Add” (supplementary file). Usually, supplementary data and tables are progressive. It is better writing “Table 6” in supplementary data. Change also in line 598
Line 354 to 358: write in better English. Do you mean…” Indirect immunofluorescence estimated by Hep2 cells showed as antinuclear factors were granular and centromeric, although fluorescence signals were variably detected also in the cytoplasm.
Line 361: correct “nucleosomeat concentrations” in “nucleosome at concentrations”
Line 363: please, introduce: (Table 5 and Fig 2)
Table 5: missing table caption
Figure 2: missing figure caption
Line 374 to 377: CENP-B and Ribosomal P0 are typical SSc antibodies, or are common antibodies present also in other autoimmune diseases?
Line 388: add description of abbreviation of genes for example: LY96 (Lymphocyte Antigen 96)
Figure 3: description is not clear and where is asterisk? Where are 120 genes on table n. 2? On the table there are 67 genes. What do you mean? On the contrary in the text the authors speak about of variants in a panel of 14 gene…
Line 388 to 399: it is not clear, please write again. In nine of fourteen patients, two or three variants of SAMD9L gene were detected and were classified as pathogenic according to the ACMG criteria? (line 389 to 391 and line 398 to 399). What do you mean as “main group” (9/14?) Moreover, describe, as for the other genes’ variation listed after, its possible role.
Line 404 to 405: Twisted sentence. Correct and write better
Line 419: add explanation that miR-146a variants can influence disease susceptibility by Affecting the activity of proteins like IRAK1, IRAK2, and TRAF6, which are crucial in signaling pathways like TNF and NF-κB
Line 426: (1/14?)
Line 430: “a single variant in the AIRE B gene was found main group (in 9 patients). Please add the %
Line 449: Please add %
Line 457: do you mean 3/9 (Main group)? Please add %
Line 467: do you mean 2/9? Please add %
Line 474: Add %
Line 476 and line 477: 3/9? Please Add %
Line 524: introduce description of abbreviations
Line 553. Missing space between [] and This
Line 574; please finish the phase “in our study we found ….and ITGA2B may be involved in SSc-ILD. Add space between ITGA2B and Given
Comments on the Quality of English LanguageEnglish may be improved in some part of article.
Author Response
Please find the response to Reviewer #1 in the uploaded file

Reviewer 2 Report
Comments and Suggestions for Authors
Thank you for the opportunity to review the manuscript titled “Systemic Sclerosis with Interstitial Lung Disease: Identification of Novel Immunogenetic Markers and Ethnic Specificity in Kazakh Patients.”
The article is highly interesting and addresses a relevant and timely topic: the immunogenetic characterization of interstitial lung disease associated with systemic sclerosis (SSc-ILD) in a population that is underrepresented in the scientific literature. Despite its potential and the use of advanced genomic technologies, the manuscript requires substantial revisions before it can be considered for publication. Below, I provide my detailed observations:
- Originality and Relevance
The focus on a Kazakh cohort represents a novel and potentially valuable contribution. However, the manuscript lacks a clear contextual explanation of the genetic, environmental, or clinical characteristics that make this population unique or particularly relevant for the study of SSc-ILD.
- Objectives and Justification
While the general objectives are understandable, they should be reformulated in more specific and structured terms. Moreover, the research hypothesis is not explicitly stated, which limits the interpretability and depth of the findings.
- Methodological Design
The small sample size (n = 30) significantly limits the statistical power of the study. This should be clearly acknowledged in the limitations section.
It is also unclear whether any statistical adjustment for multiple comparisons was applied. The absence of such correction could compromise the validity of some of the reported associations.
- Discussion
The discussion is comprehensive, but in some parts becomes redundant and lacks critical reflection.
There is no explicit acknowledgment of key methodological limitations, particularly the small sample size, absence of functional replication, and potential selection bias.
Several statements are not supported by up-to-date references, and others are expressed with a degree of certainty that is not consistent with the level of evidence presented..
Overall Assessment
The manuscript provides valuable insights and opens an important line of research within the field of precision medicine in autoimmune diseases. However, it requires significant restructuring in terms of methodology, statistical analysis, writing clarity, and critical interpretation to meet the standards expected for publication in a peer-reviewed journal.
Author Response
Please find the response to Reviewer #2 in the uploaded file
